# Stress hyperglycemia ratio predicts adverse outcomes in emergency department patients with upper gastrointestinal bleeding

Hao-Cho Ou[1], Yu-Chin An[1], Chang-Chih Shih[2], Chung-Yu Lai[3], Hui-Hsun Chiang[4], Wu-Chien Chien[5,6], Wen-I. Liao[1], Shih-Hung Tsai[1,7,8]*

1 Department of Emergency Medicine, Tri-Service General Hospital, National Defense Medical University, Taipei, Taiwan, 2 Department of Emergency Medicine, Wei Gong Memorial Hospital, Miaoli County, Taiwan, 3 Graduate Institute of Aerospace and Undersea Medicine, National Defense Medical University, Taipei, Taiwan, 4 School of Nursing, National Defense Medical University, Taipei, Taiwan, 5 School of Public Health, National Defense Medical University, Taipei, Taiwan, 6 Department of Medical Research, Tri-Service General Hospital, National Defense Medical University, Taipei, Taiwan, 7 Department of Physiology and Biophysics, Graduate Institute of Physiology, National Defense Medical University, Taipei, Taiwan, 8 Taichung Armed Forces General Hospital, Taichung, Taiwan

* tsaishihung@yahoo.com.tw

## Abstract

### Introduction

Upper gastrointestinal bleeding (UGIB) is a common emergency condition with substantial morbidity. Early identification of patients at risk for adverse outcomes is essential for timely management. The stress hyperglycemia ratio (SHR) adjusts admission glucose for baseline glycemic control and may better reflect acute physiological stress than absolute glucose levels. We aimed to determine whether SHR predicts critical outcomes in emergency department (ED) patients with UGIB.

### Methods

We retrospectively analyzed 345 adults with endoscopically confirmed UGIB at a tertiary medical center. SHR was computed as admission glucose divided by estimated average glucose from hemoglobin A1c. Multivariable logistic regression assessed associations between SHR and outcomes: intensive care unit (ICU) admission, blood transfusion, rebleeding, acute kidney injury (AKI), acute respiratory failure (ARF), in-hospital mortality, procedural intervention, and esophageal variceal (EV) bleeding. Predictive performance was compared with the complete Rockall score (CRS) and Glasgow-Blatchford score using receiver operating characteristic curves.

### Results

Elevated SHR was independently associated with higher risks of ICU admission (adjusted odds ratio [aOR] = 2.10, P < 0.001), transfusion (aOR = 6.30, P < 0.001),

**Data availability statement:** Data cannot be shared publicly because of institutional and ethical restrictions. The data underlying this study are available from the Institutional Review Board of Tri-Service General Hospital (contact via https://www.tsgh.ndmctsgh.edu.tw/) for researchers who meet the criteria for access to confidential data.

**Funding:** The author(s) received no specific funding for this work.

**Competing interests:** The authors have declared that no competing interests exist.

rebleeding (aOR = 1.75, P = 0.04), AKI (aOR = 1.79, P < 0.001), and ARF (aOR = 1.96, P = 0.01). SHR moderately predicted transfusion (area under the curve [AUC] = 0.716) and ICU admission (AUC = 0.637), outperforming the CRS for both. Adding SHR to CRS improved transfusion prediction (ΔAUC = 7.2%, P = 0.02). Patients with SHR > 1.9 had significantly higher rates of ICU admission, transfusion, organ dysfunction, and EV bleeding. SHR remained predictive in both diabetic and nondiabetic subgroups. No significant association was observed between SHR and mortality or procedural intervention.

## Conclusion

SHR was independently associated with adverse outcomes in UGIB, especially ICU admission and transfusion. As a simple, rapidly available marker that adjusts for baseline glycemic control, it may complement existing risk scores and support early, pre-endoscopic risk stratification in the ED, and warrants validation in prospective studies.

## Introduction

Upper gastrointestinal bleeding (UGIB) is a frequent and potentially life-threatening emergency, with an incidence of approximately 65 per 100,000 in the United States and substantial morbidity and mortality worldwide [1]. Diabetes mellitus (DM), affecting more than 500 million people globally [2], is present in over 10% of patients with UGIB and is associated with longer hospitalization and higher mortality [3]. UGIB often leads to complications such as acute kidney injury (AKI), acute respiratory failure (ARF), transfusion, rebleeding, and intensive care unit (ICU) admission, all of which increase healthcare burden and worsen prognosis [4,5]. Timely identification of high-risk patients is therefore crucial to optimize management and allocate resources effectively.

Stress-induced hyperglycemia (SIH) is common in acute illness [6] but its prognostic value remains inconsistent, particularly in patients with DM [7,8], where absolute admission glucose reflects both chronic dysglycemia and acute stress responses. Glycated hemoglobin (HbA1c) permits adjustment for baseline control, enabling calculation of the glycemic gap and stress hyperglycemia ratio (SHR). By accounting for chronic glycemia, these indices have shown prognostic utility in pneumonia, stroke, heart failure, and myocardial infarction [9–13].

Although hyperglycemia has been linked to adverse outcomes in UGIB [14–16], prior studies did not incorporate HbA1c-based indices, potentially underestimating the prognostic significance of stress hyperglycemia. The SHR addresses this limitation but has not been systematically evaluated in UGIB. Conventional scores such as the Glasgow-Blatchford Score (GBS) and complete Rockall Score (CRS) are widely used for risk stratification, yet GBS may overestimate risk while CRS requires endoscopic data, limiting its pre-endoscopic utility [17–19].

We therefore aimed to evaluate the prognostic value of SHR in emergency department (ED) patients with UGIB and to determine whether it provides incremental predictive performance beyond conventional scores. We hypothesized that higher SHR would be independently associated with critical outcomes including transfusion, ICU admission, organ dysfunction, and rebleeding.

## Materials and methods

### Patients

This retrospective observational study was conducted at a tertiary referral medical center in northern Taiwan and approved by the institutional review board (IRB No.: B202205204), with informed consent waived. Eligible cases were retrieved from the hospital data center using ICD-10 codes for UGIB (e.g., K92.0, K92.2, K25.0, K31.811) between 01/01/2018 and 31/12/2020 and were limited to patients with both a recent HbA1c measurement (within two months before or after admission) and an upper endoscopy performed during the index visit. Patients were included if they presented to the ED with hematemesis, melena, or both, and had endoscopic confirmation of UGIB. Of 505 encounters initially identified, 345 patients met these criteria and were included in the final analysis; the detailed selection and exclusion process is illustrated in Fig 1. Because the data center only released cases with both endoscopy and HbA1c available, all variables required for calculating the SHR (glucose and HbA1c) were complete, and no imputation was necessary. The data were accessed for research purposes between 01/03/2023 and 31/05/2023 under IRB approval, and only de-identified records were analyzed.

### Data collection

Electronic medical records were reviewed for demographics, vital signs, comorbidities, initial ED laboratory values (including glucose, hemoglobin, creatinine, platelets, and coagulation profile), total blood transfusion units, and outcomes. UGIB

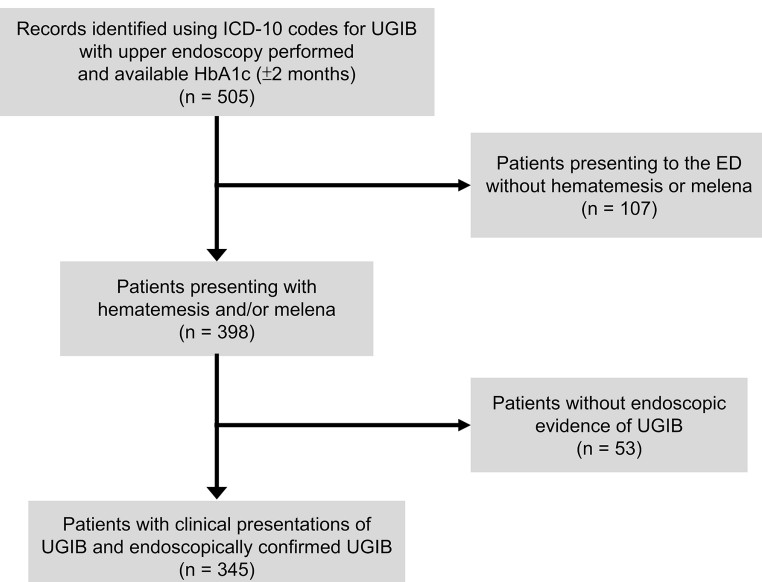

**Fig 1. Patient selection flowchart.** A total of 505 encounters were identified using ICD-10 codes for upper gastrointestinal bleeding (UGIB) among patients who had undergone upper endoscopy and had an available glycated hemoglobin (HbA1c) measurement within ±2 months of admission. Of these, 107 encounters were excluded because the emergency department (ED) presentation did not include hematemesis or melena. Among the remaining 398 patients presenting with hematemesis and/or melena, 53 were further excluded because no endoscopic evidence of UGIB was found. The final study cohort consisted of 345 patients with endoscopically confirmed UGIB and available HbA1c data.

complications included acute kidney injury (AKI, per KDIGO), acute respiratory failure (ARF, requiring ventilatory support), acute myocardial infarction (AMI, per Fourth Universal Definition), and rebleeding (defined as recurrent hematemesis or melena with shock or hemoglobin drop ≥2 g/dL after initial hemostasis) [20]. Interventions included endoscopic or surgical hemostasis. A diagnosis of DM was based on a recorded ICD code, use of hypoglycemic agents, or HbA1c ≥ 6.5%. GBS and Rockall scores were calculated using standard criteria [18,21].

## Measurements

Admission glucose was defined as the first glucose value recorded in the ED. HbA1c was measured via high-performance liquid chromatography (Primus CLC 385). Estimated average glucose (eAG) was calculated using the formula: eAG = (28.7 × HbA1c) − 46.7 [22]. The glycemic gap was defined as admission glucose minus eAG, and the SHR as admission glucose divided by eAG.

## Statistical analysis

Continuous variables were assessed for normality using the Shapiro–Wilk test. As most variables were non-normally distributed, they are presented as median with interquartile range (IQR). Categorical variables are expressed as number and percentage. Logistic regression was performed to identify independent predictors of outcomes, with results reported as unadjusted and adjusted odds ratios (aOR) with 95% confidence intervals. Covariates (age, sex, hepatic disease, cardiac failure, and diabetes mellitus) were prespecified based on clinical relevance and prior literature to minimize confounding. The optimal cutoff value of SHR for ICU admission was derived from the receiver operating characteristic (ROC) curve using the Youden index (cutoff = 1.9), which yielded a sensitivity of 46.4% and a specificity of 81.0%. Prespecified subgroup analyses were conducted after stratification by diabetes status.

Predictive performance was assessed using ROC curves and the area under the curve (AUC). The incremental value of SHR was evaluated by comparing predicted probabilities from combined logistic models (e.g., score + SHR) with those from the original scores; differences in AUCs (ΔAUC) were compared pairwise using the DeLong test. Correlations between SHR, risk scores, and transfusion volume were examined using Spearman's correlation. Missing BUN values (34 cases, 9.9%) were handled by complete case analysis; sensitivity analyses imputing normal values (0 points) yielded consistent results. With a sample size of 345, post-hoc power analysis indicated >95% power to detect the observed effect of SHR on transfusion (α = 0.05). A two-sided P < 0.05 was considered statistically significant. All analyses were performed using SPSS Statistics, version 26.0 (IBM Corp., Armonk, NY, USA).

## Results

### Baseline characteristics

A total of 345 patients were included, with a median age of 70 years and 67.5% male. Diabetes was present in 65% and chronic kidney disease in 28%. The median admission glucose was 185 mg/dL and the median SHR was 1.39. Baseline characteristics are detailed in Table 1.

### Association of SHR with outcomes

Multivariable regression confirmed that higher SHR was independently associated with ICU admission, transfusion, rebleeding, AKI, and ARF (Table 2). The strongest effect was observed for transfusion (aOR = 6.30, P < 0.001).

### Cutoff analysis

Using an SHR cutoff of 1.9, patients in the high-SHR group had significantly higher risks of ICU admission, transfusion, AKI, ARF, and esophageal variceal (EV) bleeding, while mortality and intervention rates were similar (Table 3).

**Table 1. Baseline characteristics of patients with upper gastrointestinal bleeding.**

| Variable | | All patients (n = 345) |
|---|---|---|
| Demographics | | |
| | Age | 70 (61–79.5) |
| | Male | 233 (67.5%) |
| Comorbidities | | |
| | Hepatic disease | 83 (24.1%) |
| | Cardiac failure | 17 (4.9%) |
| | Chronic kidney disease | 98 (28.4%) |
| | Diabetes mellitus | 225 (65.2%) |
| | ICU admission | 56 (16.2%) |
| Laboratory Parameters | | |
| | Admission Glucose (mg/dL) | 185 (136–243) |
| | HbA1c (%) | 6.0 (5.5–6.8) |
| | Glycemic gap (mg/dL) | 55.2 (10.5–108.8) |
| | Stress hyperglycemia ratio | 1.39 (1.09–1.87) |
| | Hemoglobin (g/dL) | 8.3 (6.5–10.1) |
| Clinical outcomes | | |
| | Blood transfusion | 304 (88.1%) |
| | Intervention | 89 (25.8%) |
| | Acute kidney injury | 97 (28.1%) |
| | Acute myocardial infarction | 12 (3.5%) |
| | Acute respiratory failure | 30 (8.7%) |
| | Rebleeding | 12 (3.5%) |
| | In-hospital mortality | 8 (2.3%) |
| | Hospital stay (days) | 6 (5–10) |

Continuous variables are presented as median (interquartile range), and categorical variables as number (percentage).

Abbreviations: ICU, intensive care unit; HbA1c, glycated hemoglobin A1c.

**Table 2. Association between stress hyperglycemia ratio and adverse outcomes in patients with upper gastrointestinal bleeding.**

| Clinical Outcome | Unadjusted OR (95% CI) | P-value | Adjusted OR (95% CI) | P-value |
|---|---|---|---|---|
| Blood transfusion | 5.16 (2.28–11.65) | < 0.001* | 6.30 (2.58–15.37) | < 0.001* |
| ICU admission | 2.08 (1.40–3.08) | < 0.001* | 2.10 (1.38–3.19) | < 0.001* |
| Rebleeding | 1.75 (1.09–2.80) | 0.02* | 1.75 (1.02–3.00) | 0.04* |
| Intervention | 1.19 (0.87–1.63) | 0.27 | 1.07 (0.77–1.48) | 0.69 |
| Mortality | 0.34 (0.07–1.59) | 0.17 | 0.33 (0.07–1.49) | 0.15 |
| AKI | 1.61 (1.15–2.25) | < 0.001* | 1.79 (1.25–2.57) | < 0.001* |
| AMI | 0.97 (0.43–2.19) | 0.95 | 1.27 (0.52–3.10) | 0.61 |
| ARF | 1.84 (1.22–2.78) | < 0.001* | 1.96 (1.17–3.29) | 0.01* |

Odds ratios (OR) and 95% confidence intervals (CI) are presented. Adjusted models included age, sex, hepatic disease, cardiac failure, and diabetes mellitus.

Abbreviations: ICU, intensive care unit; AKI, acute kidney injury; AMI, acute myocardial infarction; ARF, acute respiratory failure.

*P < 0.05.

**Table 3. Clinical outcomes stratified by stress hyperglycemia ratio using a cutoff value of 1.9.**

|  | SHR ≤ 1.9 (n = 265) | SHR > 1.9 (n = 80) | *P*-value |
|---|---|---|---|
| Blood transfusion | 224 (84.5%) | 80 (100.0%) | < 0.001* |
| ICU admission | 30 (11.3%) | 26 (32.5%) | < 0.001* |
| Rebleeding | 6 (2.3%) | 6 (7.5%) | 0.10 |
| Intervention | 65 (24.6%) | 24 (30.0%) | 0.34 |
| Mortality | 7 (2.6%) | 1 (1.3%) | 0.47 |
| AKI | 65 (24.5%) | 32 (40.0%) | < 0.001* |
| AMI | 9 (3.4%) | 4 (3.8%) | 0.88 |
| ARF | 17 (6.4%) | 13 (16.3%) | 0.03* |
| EV bleeding | 23 (8.7%) | 20 (25.0%) | < 0.001* |
| GI malignancy | 16 (6.1%) | 7 (8.8%) | 0.40 |

Values are presented as number (percentage). P-values were calculated using the Chi-square test or Fisher's exact test, as appropriate.

Abbreviations: ICU, intensive care unit; AKI, acute kidney injury; AMI, acute myocardial infarction; ARF, acute respiratory failure; EV, esophageal varices; GI, gastrointestinal.

*$P < 0.05$.

### Correlation analysis

SHR showed a positive linear correlation with both the GBS and the CRS, suggesting it reflects similar risk dimensions as traditional UGIB scoring systems. A moderate positive correlation was also observed between SHR and the total volume of red blood cell transfusion (rho = 0.278, P < 0.001). These relationships are illustrated in Fig 2.

### Predictive performance

SHR demonstrated moderate discriminative ability for transfusion (AUC = 0.716) and ICU admission (AUC = 0.637), out-performing the CRS. Adding SHR significantly improved the CRS for predicting transfusion (ΔAUC = 7.2%, P = 0.02), while improvements for other outcomes were modest (Fig 3, Table 4).

### Subgroup analysis

Subgroup analyses stratified by diabetes status demonstrated broadly consistent associations between higher SHR and adverse outcomes (Supplementary S1 Table). Among patients with diabetes, SHR was independently associated with ICU admission, transfusion, and acute kidney injury. In nondiabetic patients, SHR significantly predicted transfusion, rebleeding, and acute respiratory failure. These findings suggest that SHR retains prognostic utility regardless of baseline diabetic status.

### Discussion

In this retrospective cohort of patients with UGIB, we found that the SHR, a metric incorporating both acute and chronic glycemic status, was independently associated with several adverse clinical outcomes. Higher SHR predicted ICU admission, transfusion requirement, AKI, ARF, and rebleeding, and it also showed discriminatory ability for EV bleeding. Among these, its predictive value was most robust for transfusion and ICU admission, where it outperformed the Rockall score. Importantly, these associations remained consistent in patients with and without diabetes, highlighting SHR as a clinically useful marker beyond pre-existing glycemic status.

SIH has been recognized as a marker of physiological stress and adverse outcomes across various acute illnesses [23–26]. However, its prognostic role in UGIB has remained unclear. Existing risk scores such as the GBS and Rockall

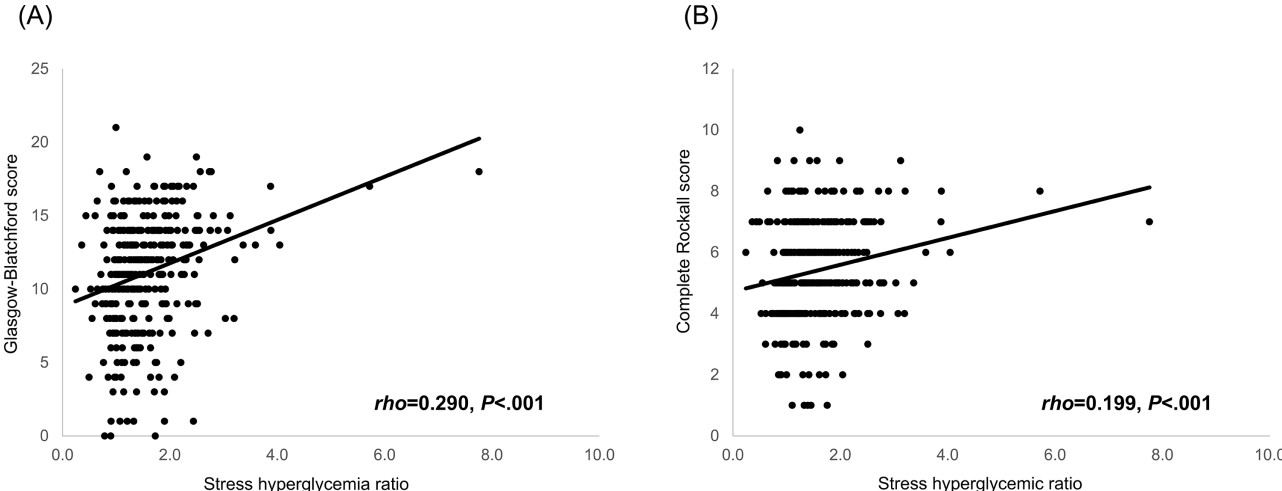

**Fig 2. Correlation of the stress hyperglycemia ratio with established risk scores.** Scatterplots demonstrate the relationship between the stress hyperglycemia ratio and two prognostic scores: (A) Glasgow-Blatchford score and (B) complete Rockall score. Each dot represents an individual patient, and the solid line indicates the fitted regression line.

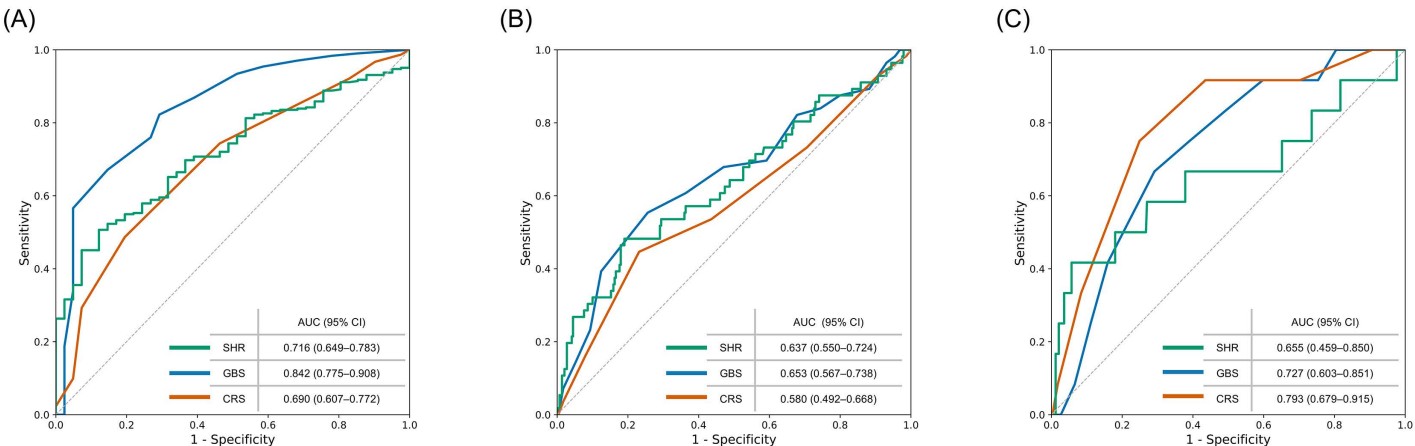

**Fig 3. Receiver operating characteristic curves for predicting clinical outcomes.** Receiver operating characteristic (ROC) analyses were performed to evaluate the discriminatory ability of the stress hyperglycemia ratio (SHR), the Glasgow-Blatchford score (GBS), and the complete Rockall score (CRS). (A) ROC curve for predicting blood transfusion. (B) ROC curve for predicting intensive care unit admission. (C) ROC curve for predicting rebleeding. The diagonal line represents the reference line of no discrimination.

score are widely used but have notable limitations, including lack of specificity or reliance on endoscopic findings [27]. Unlike the glycemic gap, the SHR standardizes acute glycemic elevation relative to baseline control, thereby minimizing interpatient variability [28,29]. In our study, SHR demonstrated stronger and more consistent predictive performance than the glycemic gap, supporting its utility as a simple, early marker of severity in UGIB.

Predicting ICU admission in UGIB is clinically important for management and resource allocation. Previous studies have shown that GBS and the Rockall score can predict ICU admission, with AUC values comparable to ours [30,31]. However, these scores have limitations. For example, Yang et al. proposed a model with good accuracy (AUC = 0.850) but excluded glycemic parameters, and although their ICU patients had higher glucose levels, the difference was not

**Table 4. Receiver operating characteristic analysis of stress hyperglycemia ratio combined with conventional risk scores for predicting major clinical outcomes.**

| Model | AUC (95% CI) | *P*-value | ΔAUC (%, 95% CI) | *P*-value |
|---|---|---|---|---|
| ICU admission | | | | |
| CRS + SHR | 0.645 (0.561–0.730) | < 0.001* | 6.5 (−1.8–14.8) | 0.13 |
| GBS + SHR | 0.670 (0.586–0.754) | < 0.001* | 1.7 (−3.7–7.1) | 0.53 |
| Rebleeding | | | | |
| CRS + SHR | 0.824 (0.708–0.939) | < 0.001* | 3.1 (−1.7–7.8) | 0.20 |
| GBS + SHR | 0.760 (0.627–0.893) | < 0.001* | 3.3 (−2.8–9.3) | 0.29 |
| Blood transfusion | | | | |
| CRS + SHR | 0.762 (0.688–0.836) | < 0.001* | 7.2 (1.2–13.2) | 0.02* |
| GBS + SHR | 0.860 (0.797–0.922) | < 0.001* | 1.8 (−1.0–4.5) | 0.21 |

Values are presented as the area under the receiver operating characteristic curve (AUC, expressed as decimals) with 95% confidence intervals (CI). ΔAUC represents the change in AUC after adding the stress hyperglycemia ratio (SHR) to each risk score and is expressed as a percentage, with statistical significance assessed using the DeLong test.

Abbreviations: CRS, complete Rockall score; GBS, Glasgow-Blatchford score; ICU, intensive care unit.

*P < 0.05.

statistically significant, likely due to lack of adjustment for baseline control [32]. In our cohort, elevated SHR was independently associated with ICU admission, aligning with findings in other acute conditions such as pneumonia and acute heart failure, where increased glycemic gap predicted higher ICU risk [9,23].

Blood transfusion remains a critical intervention in UGIB. The GBS, designed by Blatchford et al., was specifically aimed at predicting transfusion and other interventions using non-endoscopic variables [18], and correlates strongly with transfusion volume (r = 0.922, P < 0.001) [33]. Comparative studies consistently show GBS outperforms Rockall in transfusion prediction—for example, Custovic et al. reported higher accuracy for GBS (AUC = 0.810) versus Rockall (AUC = 0.675) [34]. Consistent with these findings, SHR emerged as a novel independent predictor of transfusion in our study. Elevated SHR was significantly associated with transfusion (aOR = 6.30; P < 0.001), and 100% of patients with SHR > 1.9 required transfusion. SHR also correlated positively with transfusion volume, supporting its role as a marker of bleeding severity. ROC analysis further showed SHR outperformed the CRS, and adding SHR significantly improved CRS prediction (ΔAUC = 7.2%, P = 0.02). Although combining SHR with GBS increased AUC without statistical significance (ΔAUC = 1.8%, P = 0.21), these findings suggest that SHR provides incremental prognostic value and may serve as a useful adjunct to established scores for guiding timely transfusion decisions. Multiple physiological mechanisms could potentially explain the relationship between SHR and transfusion. Acute hemorrhage elicits a robust sympathetic and hypothalamic–pituitary–adrenal (HPA) axis response, leading to catecholamine- and cortisol-driven increases in hepatic glucose output and insulin resistance [35]. Patients requiring transfusion typically experience greater hemodynamic stress, producing proportionally higher stress hyperglycemia. Thus, SHR likely reflects the neuroendocrine stress response to bleeding severity, providing a physiologically plausible explanation for its association with transfusion.

AKI in UGIB patients has been linked to higher mortality, longer ICU stays, and increased costs [4], and hyperglycemia has been identified as an independent predictor of AKI in this setting [36]. In our study, elevated SHR was independently associated with both ARF and AKI, with significantly higher event rates in patients with SHR > 1.9. These findings suggest that glycemic dysregulation, as captured by SHR, may better reflect systemic stress and organ vulnerability than baseline clinical variables alone. This observation is consistent with reports in other acute conditions, such as pneumonia and acute heart failure, where elevated glycemic gap has been linked to ARF [9,23], and to AKI in pneumonia and necrotizing

fasciitis [23,37]. Collectively, our results support SHR as an early metabolic stress marker to identify UGIB patients at heightened risk for end-organ injury and to guide timely initiation of organ support.

Rebleeding is a critical adverse outcome in UGIB, significantly influencing subsequent management and prognosis. Both the GBS and CRS have shown only moderate utility in predicting this risk, and prior studies highlight predictors such as hemodynamic instability, large ulcer size, and stigmata of recent hemorrhage [38]. More recently, models incorporating laboratory biomarkers such as lactate and D-dimer have improved risk stratification [39]. In our study, elevated SHR was independently associated with rebleeding (aOR = 1.75, P = 0.04) after adjustment for key covariates. Although patients with SHR > 1.9 had higher recurrence rates, the difference was not statistically significant, likely due to the low event rate. These findings suggest that glycemic dysregulation captured by SHR may contribute to rebleeding risk and may complement existing scores in identifying high-risk patients, although the incremental predictive value was modest and did not reach statistical significance in our analysis. The association between SHR and rebleeding may be explained by complementary hemodynamic and biological mechanisms. As with its relationship to transfusion, higher SHR likely reflects more severe initial hemorrhage, and the severity of the initial bleed is itself a recognized determinant of rebleeding in UGIB [40]. In addition, stress hyperglycemia is linked to inflammatory activation and microvascular dysfunction, which can impair mucosal healing and clot stability, further predisposing to recurrent bleeding [41]. Together, these pathways suggest that SHR serves as an integrated marker of both initial bleeding severity and impaired healing responses, providing a biologically plausible explanation for its association with rebleeding.

EV bleeding differs fundamentally from non-variceal bleeding in pathophysiology and prognosis, as it arises from portal hypertension and often presents with massive hemorrhage in cirrhotic patients, carrying higher risks of rebleeding and mortality [42,43]. In our study, SHR demonstrated better predictive performance for EV bleeding compared with GBS (AUC = 0.680 vs. 0.655) and both pre- and post-endoscopy Rockall scores (AUC = 0.496 and 0.595), supporting its potential role as an adjunctive marker in this high-risk group. This association may reflect the unique metabolic profile of cirrhosis, characterized by impaired insulin sensitivity and altered glucose utilization [44]. Nearly all EV bleeding patients in our cohort had underlying hepatic disease, which likely contributed to impaired baseline glycemic control and exaggerated stress hyperglycemia. These findings suggest that SHR may capture the dual metabolic burden in cirrhotic patients and offer incremental value for risk stratification in EV bleeding.

SHR was not significantly associated with in-hospital mortality (aOR = 0.33; P = 0.15) or the need for endoscopic or surgical intervention (aOR = 1.07; P = 0.69), with comparable event rates across SHR groups. These findings align with prior studies in pneumonia, liver abscess, and necrotizing fasciitis [23,37,45], where stress-related glycemic markers correlated with disease severity but did not consistently predict mortality. This suggests that while SHR reflects acute physiological stress, ultimate outcomes such as death or procedural need are multifactorial and may not be captured by glycemic markers alone. To further explore the lack of association between SHR and overall mortality, we additionally compared major treatment-related variables between survivors and non-survivors. Transfusion rates and intervention frequencies did not differ significantly, and although ICU admission was more common among non-survivors, ICU triage in our institution is driven by clinical deterioration rather than blood glucose levels. These findings suggest that treatment intensity was not disproportionately escalated based on hyperglycemia and is therefore unlikely to explain the absence of a mortality association. In contrast, non-survivors demonstrated markedly higher rates of acute organ dysfunction and pre-existing comorbidity, indicating that mortality in UGIB is predominantly influenced by systemic decompensation rather than glycemic dysregulation. Given the very low mortality rate in our cohort (2.3%), limited statistical power may also have contributed to the absence of an independent association between SHR and mortality.

This study has several strengths. As a simple and rapidly obtainable parameter, SHR offers meaningful clinical value for early risk stratification in patients with UGIB. By quantifying acute physiological stress relative to baseline glycemic control, SHR serves as a standardized and objective marker that complements—but does not replace—traditional risk scores. In our cohort, SHR effectively identified patients at higher risk of ICU admission, transfusion, ARF, and AKI, despite limited utility for mortality or procedural prediction. Its ease of calculation and immediate availability before endoscopy highlight its potential as a practical adjunct to support triage and prioritize care in the emergency setting.

In interpreting SHR, it is important to consider factors that may affect HbA1c, which is used to estimate baseline glycemia. HbA1c can be influenced by anemia through changes in erythrocyte lifespan, with chronic iron-deficiency anemia potentially elevating values and acute blood loss or transfusion lowering them independent of glycemia [46]. These mechanisms may introduce some measurement variability. However, large population-based analyses indicate that hemoglobin concentration itself has only a minimal effect on HbA1c, accounting for an absolute difference of approximately 0.2% across the full range of hemoglobin values [47]. Therefore, the acute reductions in hemoglobin commonly observed in UGIB are unlikely to meaningfully distort HbA1c values or substantially bias SHR calculation in this cohort.

Several limitations should be acknowledged. This was a retrospective, single-center analysis from a tertiary referral hospital in Taiwan, which may limit generalizability due to selection bias and practice variation. The requirement for recent HbA1c data could have further introduced bias, as patients with available measurements may differ systematically from those without. Moreover, the low incidence of certain outcomes—particularly mortality (2.3%) and rebleeding (3.5%)—reduced statistical power, and non-significant findings should therefore be interpreted with caution. Finally, both the predictive models and the optimal SHR cutoff (1.9) were derived from a single dataset without external validation. Prospective, multicenter studies across diverse populations are warranted to confirm reproducibility and to determine how SHR can be optimally integrated into clinical pathways for risk stratification in UGIB.

## Conclusion

SHR was independently associated with adverse outcomes in UGIB, particularly transfusion and ICU admission, and outperformed the CRS in these settings. As a simple and rapidly available marker that accounts for baseline glycemic control, SHR may complement conventional risk scores and support early, pre-endoscopic risk stratification in the emergency department. These findings warrant confirmation in future prospective, multicenter studies.

## Supporting information

**S1 Table. Association between stress hyperglycemia ratio and adverse outcomes in patients with and without diabetes mellitus.** Unadjusted and adjusted odds ratios (ORs) with 95% confidence intervals (CIs) were calculated using logistic regression analyses. Multivariable models were adjusted for age, sex, hepatic disease, and chronic heart failure. ICU, intensive care unit; AKI, acute kidney injury; AMI, acute myocardial infarction; ARF, acute respiratory failure. *P<0.05.
(DOCX)

## Acknowledgments

The authors thank the Data Center of Tri-Service General Hospital for providing data support for this study.

## Author contributions

**Conceptualization:** Shih-Hung Tsai, Chang-Chih Shih, Wen-I Liao.

**Data curation:** Hao-Cho Ou, Chang-Chih Shih.

**Formal analysis:** Hao-Cho Ou, Chung-Yu Lai, Hui-Hsun Chiang, Wu-Chien Chien.

**Funding acquisition:** Hao-Cho Ou.

**Methodology:** Hao-Cho Ou, Hui-Hsun Chiang.

**Supervision:** Shih-Hung Tsai, Hui-Hsun Chiang, Wen-I Liao.

**Writing – original draft:** Hao-Cho Ou.

**Writing – review & editing:** Hao-Cho Ou, Shih-Hung Tsai, Yu-Chin An.

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
