## [Decision Letter · Decision Letter 0]

30 Nov 2025

Dear Dr. Tsai,

Thank you for submitting your manuscript to PLOS ONE. After careful consideration, we feel that it has merit but does not fully meet PLOS ONE’s publication criteria as it currently stands. Therefore, we invite you to submit a revised version of the manuscript that addresses the points raised during the review process.

**ACADEMIC EDITOR:  Major revision**

We look forward to receiving your revised manuscript.

Kind regards,

Marwan Salih Al-Nimer, MD, PhD

Academic Editor

PLOS ONE

Journal Requirements:

Additional Editor Comments:

This single center, retrospective study need clarification for the following comments.

#1 typing errors

#2 The authors adjust the SHR> 1.9, add the sensitivity and specificity for this cut off value because we cant calculated Youden's index without sensitivity and specificity

#4: Add reference for the eAG formula

#5: How the author deals with missing data. Flow chart is necessary to outline the process of methodology

#6: The tables showed that the continuous data are not normally distributed. Therefore the results mean +/- SD, as well as the analysis will bias the conclusion

#7: Tables are not clear (the title, and footnotes)

#8: Improve the production of Figures

#9: update the references as >40% are old references.

Reviewers' comments:

Reviewer's Responses to Questions

**Comments to the Author**

1. Is the manuscript technically sound, and do the data support the conclusions?

Reviewer #1: Partly

2. Has the statistical analysis been performed appropriately and rigorously?

Reviewer #1: Yes

3. Have the authors made all data underlying the findings in their manuscript fully available?

Reviewer #1: No

4. Is the manuscript presented in an intelligible fashion and written in standard English?

Reviewer #1: Yes

Reviewer #1: In this manuscript, Ou et al. examine the relationship between stress hyperglycemia – the degree of glycemic variation from the patient’s baseline glucose – and outcomes in patients who present with upper gastrointestinal bleeding (UGIB), and demonstrate an independent association of stress hyperglycemia with multiple adverse outcomes. In my view this is an extremely important study, and while novelty is not required at PLOS journals, there is real novelty in this study’s identification of stress hyperglycemia – as opposed to baseline glycemic control – as a predictor of outcomes. My comments are minor and can be addressed textually:

1. It would be helpful for the reader to speculate on potential mechanistic associations between transfusion and stress hyperglycemia, and similarly between rebleeding and stress hyperglycemia; while this is not the goal of the study, it would help place the interesting findings in context.

2. Is it possible that stress hyperglycemia does not correlate with overall mortality because hyperglycemia triggers a more intensive treatment regiment? It would be helpful to comment on how local treatment algorithms are affected by blood glucose.

3. Baseline glycemic control is determined by A1c, which is perfectly reasonable, but it’s well known that anemia can affect A1c. This appears to be a confounder with the relationship between glycemic control, UGIB, and rebleeding. Please consider commenting on this.

**Do you want your identity to be public for this peer review?** For information about this choice, including consent withdrawal, please see our Privacy Policy

Reviewer #1: No

---

## [Author Response · Author response to Decision Letter 1]

19 Dec 2025

We thank the editor and reviewer for their constructive comments on our manuscript.

All comments have been addressed point by point in the attached “Response_to_Reviewers” document.

All revisions are highlighted in the revised manuscript with track changes.

---

## [Editor Report · Decision Letter 1]

22 Dec 2025

Stress hyperglycemia ratio predicts adverse outcomes in emergency department patients with upper gastrointestinal bleeding

PONE-D-25-51490R1

Dear Dr., Shih Hung Tsai

We’re pleased to inform you that your manuscript has been judged scientifically suitable for publication and will be formally accepted for publication once it meets all outstanding technical requirements.

Kind regards,

Marwan Salih Al-Nimer, MD, PhD

Academic Editor

PLOS One
---

## [Editor Report · Acceptance letter]

PONE-D-25-51490R1

PLOS One

Dear Dr. Tsai,

I'm pleased to inform you that your manuscript has been deemed suitable for publication in PLOS One. Congratulations! Your manuscript is now being handed over to our production team.

Kind regards,

on behalf of

Professor Marwan Salih Al-Nimer

Academic Editor

PLOS One